# The Presence of *Toxocara* Eggs on Dog’s Fur as Potential Zoonotic Risk in Animal-Assisted Interventions: A Systematic Review

**DOI:** 10.3390/ani9100827

**Published:** 2019-10-19

**Authors:** Maria Paola Maurelli, Antonio Santaniello, Alessandro Fioretti, Giuseppe Cringoli, Laura Rinaldi, Lucia Francesca Menna

**Affiliations:** 1Unit of Parasitology and Parasitic Diseases, Department of Veterinary Medicine and Animal Production, University of Naples Federico II, Via Delpino 1, 80137 Napoli, Italy; giuseppe.cringoli@unina.it (G.C.); laura.rinaldi@unina.it (L.R.); 2Unit of Infectious Diseases of Domestic Animals, Department of Veterinary Medicine and Animal Production, University of Naples Federico II, via Mezzocannone, 8-80134 Napoli, Italy; alessandro.fioretti@unina.it (A.F.); luciafrancesca.menna@unina.it (L.F.M.)

**Keywords:** zoonosis, animal-assisted interventions, public health, fur, dog

## Abstract

**Simple Summary:**

Animal-assisted interventions (AAIs) represent an opportunity for the well-being and health of people, but it is necessary that the animals involved in these interventions are subjected to very thorough health checks in order to avoid the potential risk of zoonoses transmission. Dogs are the main animal species involved in AAIs and may represent a potential reservoir of zoonotic agents (e.g., bacteria, parasites, fungi). Some scientific contributions have been published regarding healthcare checks and related hygiene measures for dogs involved in these interventions, but no attention has been paid to the presence of *Toxocara* eggs on the fur. Thus, a systematic review was carried out to address this topic. Although the infection of humans through the transmission of *Toxocara* eggs after direct contact with dogs must be critically challenged, we suggest including the examination of fur during a complete parasitological screening of dogs involved in AAIs in order to exclude hair coat contamination with zoonotic helminth eggs. Moreover, it is important to also monitor the behaviors of dogs that can increase the risk of contamination from the environment (e.g., roll on grass and feces of other dogs or cats) as well as the life habits of dogs (e.g., outdoor or indoor).

**Abstract:**

Animal-assisted interventions (AAIs) usually contribute to the well-being and health of users/patients, but it is essential that the animals involved in these activities do not represent a source of zoonoses. This systematic review focused on the evaluation of the potential risk of the transmission of *Toxocara* by dogs’ fur, considering their involvement as the main animal species in AAIs. Three databases were considered: MEDLINE/PubMed, Scopus, and Web of Science, and the PRISMA guidelines were used. Out of 162 articles found, 14 papers were identified as eligible for inclusion in the review. Although the findings were very heterogeneous, they showed that regular parasitological surveillance to plan effective control programs is strongly needed to guarantee the health of pets and consequently the public health, according to the concept of One Health. Since AAIs involve patients and/or users potentially susceptible, it is very important to appropriately treat dogs enrolled in these interventions after an accurate diagnosis of parasitic zoonoses.

## 1. Introduction

Animals can contribute to many aspects of human wellbeing, health, and education through their involvement in animal-assisted interventions (AAIs) [1,2,3,4,5] that are defined as “Goal oriented and structured interventions that intentionally include or incorporate animals in health, education and human services (e.g., social work) for the purpose of therapeutic gains in humans” [6]. AAIs include human–animal teams in formal human services such as animal-assisted therapy (AAT), animal-assisted education (AAE), or animal-assisted activity (AAA). Commonly, AAIs are increasingly performed in support of healthcare within a wide range of physical and mental health problems in hospitals, rehabilitation clinics, psychiatric facilities, prisons, schools, nursing homes, etc. [7]. As reported by Shen et al. [8] and Glenk [9], various domestic animal species are involved in AAIs, but dogs are the most widely studied and most widely involved animal, especially in AAT.

In the context of AAIs, patients and/or users interact with dogs and such interaction can include several relational activities such as petting, physical contact, brushing, playing, and strolling with the dog. Particularly, it should be noted that “bodily contact” is one of the main features contributing to AAIs effectiveness, even in different settings [8]. In fact, during these activities, the patients (e.g., immunocompromised individuals, elderly, and children) and/or the users continuously come into contact with the dog (and also with its fur), thus being potentially exposed to zoonotic agents such as bacteria, fungi, and parasites [10,11,12,13] even when dogs are asymptomatic [14,15]. 

*Toxocara canis* is one of the most widespread zoonotic parasites in Europe [14]. Canids that are the definitive hosts of this parasite become infected, ingesting embryonated eggs from the environment or larvae in paratenic hosts (e.g., rodents) [16]. Puppies can also be infected vertically by transplacental or transmammary transmission from the bitch [14]. Infected definitive hosts excrete eggs of *T. canis* in the environment with feces and after a period of 2–6 weeks they become infective, depending on soil type and environmental conditions such as temperature and humidity. The eggs of *T. canis* are very resistant and can survive in the environment for months to years, under optimal circumstances [14]. 

Diagnosis of *T. canis* is traditionally performed using copromicroscopic techniques to detect eggs that can be differentiated from zoonotic *T. cati* eggs either through an accurate morphological examination [17] or by molecular analysis [18]. However, it should be noted that identification of *Toxocara* eggs to species level have seldom been reported in the scientific literature, and many papers that were published in the past used the term “*T. canis*“ for eggs of *Toxocara* spp. without exact species determination. However, it is worth underlining that some authors [17,19,20] showed the presence of *T. cati* in dog feces with a percentage ranging from 7.3 to 34.5% of *Toxocara* eggs in the feces of dogs. The presence of *T. cati* could be mainly attributed to coprophagia of cat feces by dogs [20].

Both species (*T. canis* and *T. cati*) are of zoonotic importance, causing human toxocarosis. The main route of infection for humans is by oral ingestion of embryonated eggs of *Toxocara* spp. (e.g., by food and water contaminated or unwashed hands dirty with contaminated soil) [21]. *Toxocara* spp. infections in humans cause visceral, ocular, neuronal *larva migrans*, and occult toxocarosis [22]. Visceral *larva migrans* is sometimes asymptomatic, but the common clinical signs are coughing, asthma, bronchospasm, myalgia, abdominal pain, anorexia, occasionally myocarditis, or cutaneous manifestations (e.g., pruritus, rash, eczema, etc.). Ocular *larva migrans* causes decreased vision, ophthalmitis, chorioretinitis, and unilateral or bilateral blindness. Neuronal *larva migrans* causes meningitis, encephalitis, cerebral vasculitis, or myelitis. Finally, occult toxocarosis does not present specific symptoms [23,24].

Public parks, playgrounds, sandpits, etc. may become areas of *Toxocara* infection for humans and dogs [14]. Children mainly become infected in this way, because they are frequently in contact with contaminated soil/sand and could practice geophagia [25]. In Italy, environmental contamination of *Toxocara* spp. eggs was evaluated in different cities, with a prevalence of 33.6% in the Marche region, 7.0% in Milan, 3.6% in Messina and Teramo, 2.5% in Bari, 1.9% in Rome, 0.7% in Naples and in Padua, and 0.5% in Alghero, as reviewed by Traversa et al. [26].

The prevention of zoonotic risks represents one of the main objectives of the veterinary profession, therefore, the aim of this review was to focus our attention on the potential risk of *Toxocara* eggs transmission through contact with dogs’ fur.

Considering that few data are available in the scientific literature regarding the health and welfare of animals involved in AAIs [27,28,29], our findings could be useful to promote parasite control plans for healthy dogs involved in the AAIs, encouraging the cooperation between human and veterinary medicine according to the concept of One Health [30,31,32]. 

## 2. Materials and Methods

### 2.1. Systematic Review Protocol

This review was performed according to the Preferred Reporting Items for Systematic Reviews and Meta-Analyses (PRISMA) [33]: (1) preparation of a database search to detect potentially related articles, (2) assessment of the relevancy of papers, (3) evaluation of quality, and (4) data extraction.

### 2.2. Search Strategy and Data Sources

Two researchers (A.S. and M.P.M.) independently performed the systematic search using the following strings: *Toxocara canis* AND “hair”, *Toxocara canis* AND “fur”, *Toxocara canis* AND “coat”. We also accepted all the contributions documenting *T. canis* on dog’s fur if the authors did not perform any exact species determination (i.e., *T. canis* or *T. cati*) by morphological or molecular analysis. For this reason, we chose to refer to the genus name “*Toxocara*” throughout the review.

Papers on the presence of eggs of *Toxocara* on dogs’ fur were sorted by title and abstract and then screened to remove duplicates before the final selection. Original research English studies (published or in press) were included, while reviews, comments, letters, etc. without original data were excluded from this systematic review (Appendix A). Other related papers including references from selected papers were revised and used as supplementary information sources. In our study, the scientific literature published until 31 May 2019 on the topic of this review (see below) was included using three scientific electronic databases: MEDLINE/PubMed [34], Scopus [35], and Web of Science [36].

### 2.3. Quality Assessment and Data Extraction

The papers for full text review were considered eligible if they contained information related to the risks of zoonotic transmission of *Toxocara* eggs through the fur of dogs; whereas those focusing on the epidemiology of *Toxocara* spp. based on copromicroscopic surveys were excluded. No restrictions were applied based on age, breed, health status, and living conditions of dogs, nor regarding the technique used to detect the eggs of *Toxocara*. During the first screenings, duplicate or irrelevant papers were excluded. 

The same investigators (A.S. and M.P.M.) separately assessed the papers considering the criteria enunciated above, according to the PRISMA protocol [33]. 

## 3. Results

The search returned a total of 162 documents. After removing the duplicates and irrelevant results, 49 articles were obtained for the complete revision of the entire text. Following the final evaluation, 14 sources were obtained for the systematic review (Figure 1). 

### Study Selection and Characteristics

Table 1 reports the main findings for each of the 14 studies included in this systematic review.

The 14 studies were published between 2003 and 2018. Five papers (35.7%) were performed in Europe [37,38,39,40,41], four (28.6%) in Asia [22,42,43,44], two (14.3%) in Africa [23,45], and three (21.4%) in America [46,47,48]. 

The number of dogs enrolled in these studies ranged from 51 [22] to 267 [45] with different attitudes (owned, stray/kennelled, hunting, and sheep dogs), but most of them were owned dogs (in 11/14 papers = 78.6%).

Seven papers focused on the sampling of dog fur and feces (50%) [22,23,38,40,43,44,45], whilst six only on fur sampling (42.9%) [39,41,42,46,47,48], finally one study was based on fur collection and the euthanasia of dogs (7.1%) [37]. 

Fur samples were collected only from the perianal region in one study [46], whereas in the other papers, fur samples were collected on different body regions, from two (perianal and dorsum) [37,42] to seven (head, neck, ventral and lateral abdomen, perianal, hindquarters, and tail region) [44]. 

The percentage of dogs with *Toxocara* spp. eggs on fur ranged from a minimum of 2.9% [38] to a maximum of 67.0% [37], whilst dogs with positive fecal samples for *Toxocara* spp. ranged from 4.4% [40] to 76.2% [45]. In most studies, positive fecal samples were higher than the positive fur samples excluding Overgaauw et al. [40], who revealed 4.4% of positivity in feces and 12.2% on fur. Sowemimo and Ayanniyi [45] showed that dogs positive for fur were all positive for feces. Similar results were also reported by El-Tras et al. [23] for domestic dogs, whilst for stray dogs, two animals (3.8%; CI 95% = 0.7–14.1%) showed positive hair samples, but negative fecal samples. Finally, Oge et al. [43] did not find dogs positive for fur and feces contemporaneously. 

Moreover, different risk factors (age, gender, hair length/breed, coat type, body weight, attitude of dogs, soil contact) that could influence the presence of *Toxocara* eggs on fur were evaluated in the papers analyzed. Age of the enrolled dogs ranged from 34 days [22] to 15 years [39]. In three papers [39,40,41], higher prevalence was found in adults (>12 months) than in young dogs (6 months–1 year) and puppies (<6 months) (*p* < 0.05). In contrast, other authors reported a higher prevalence of positivity for *Toxocara* spp. on fur in puppies [22,23,37,38,42,46,47]. No significant association with age was found by Oge et al. [43], Sivajothi and Reddy [44], Sowemimo and Ayanniyi [45], and Rojas et al. [48].

Regarding gender, some papers have shown higher positivities (presence of *Toxocara* eggs on dog’s fur) either in female [23,46,47] or in male [22,42,45,48] dogs; however, these authors did not report any significant difference based on gender.

No associations were found with other risk factors as body weight, coat type, and soil contact (*p* > 0.05). Only Merigueti et al. [47] found an association between positivity and half-breed dogs (*p* = 0.0099), whilst Sowemimo and Ayanniyi [45] showed that there were significant differences of positivity between local (African shepherd) and exotic breeds (*p* < 0.05).

The number of *Toxocara* eggs collected from fur ranged from 26 [41] to 39,120 [37]. Only some authors identified the species (i.e., *T. canis*) of eggs recovered from fur by morphological or molecular analysis [22,23,39,42,45,46].

Viability of eggs recovered from fur was evaluated in 11/14 (78.6%) papers according to their morphological characteristics: non-viable eggs (not intact), viable/unembryonated (intact egg with content), embryonating (egg with two or more cell divisions), and embryonated (containing *larvae*) [22,37]. Viable eggs were found by Wolfe and Wright [39], Aydenizöz-Özkayhan et al. [22], Roddie et al. [37], Amaral et al. [46], and Merigueti et al. [47] with a prevalence of 50.7%, 79.0%, 70.8%, 53.0%, and 86.6%, respectively. Embryonating eggs, instead, were found by Wolfe and Wright [39], Aydenizöz-Özkayhan et al. [22], Roddie et al. [37], Amaral et al. [46], and Merigueti et al. [47] with a prevalence of 23.9%, 12.9%, 70.8%, 2.0%, and 13.4%, respectively. Keegan and Holland [41] and Oge et al. [43] found only one embryonating egg with a prevalence of 0.7% and 3.8%, respectively. Moreover, embryonated eggs were found by Wolfe and Wright [39], Roddie et al. [37], Aydenizöz-Özkayhan et al. [22], El-Tras et al. [23] with prevalence of 4.2%, 0.3%, 8.1% and 2.4 %, respectively. Finally, Overgaauw et al. [40], Sowemimo and Ayanniyi [45], and Paoletti et al. [38] found unembryonated eggs in all fur samples analyzed.

Different authors have also evaluated the fur length and its correlation with egg viability [22,38,40,41,46,47]. Amaral et al. [46] found about 86% of the viable eggs on short hair dogs, with a difference statistically significant when compared to long hair dogs (*p* < 0.0001). Similarly, Roddie et al. [37] found a higher prevalence of embryonation rate in puppies than in adult dogs.

## 4. Discussion

The scientific literature used in this systematic review highlighted the possibility of the transmission of *Toxocara* spp. to people and other dogs through contact with the fur of dogs contaminated by eggs.

Overgaauw and von Knapen [52], Overgaauw et al. [40], Keegan and Holland [41], and Nagy et al. [19] suggested that there was a low risk of infection with eggs of *Toxocara* spp. on fur due to the low prevalence of embryonated eggs (0–4%) found in their studies. Moreover, Overgaauw et al. [40] also showed that in the case of highly contaminated fur with embryonated eggs (i.e., 12 *eggs per gram*, by Wolfe and Wright [39]), more than 4 g of hair sample was necessary to ingest 50 infective eggs.

However, Aydenizöz-Özkayhan et al. [22] found a higher number of eggs per gram of *Toxocara* on fur than in soil: 18.05 eggs per gram of hair vs. 0.09 per 30 g and 0.067 per 100 g of soil. Moreover, Oge et al. [43] showed that dog feces were negative for *Toxocara* spp., whilst the dogs’ fur samples were positive, representing a silent potential risk for humans. Therefore, although soil contamination is the main cause of most cases of *larva migrans* in humans, the transmission of *Toxocara* eggs by direct contact with dogs should not be underestimated.

The findings of the analyzed papers highlighted that the source of eggs on fur could be different for stray dogs as well as for adult dogs; in fact, they could acquire *Toxocara* through contact with contaminated environments whilst for owned dogs and puppies, it could be due to self-contamination as reported by Roddie et al. [37], El-Tras et al. [23], and Sowemimo and Ayanninyi [45]. This hypothesis was also confirmed by Roddie et al. [37], who found a positive correlation between the number of worms collected after euthanasia and the number of *Toxocara* eggs on the puppies’ fur, therefore adult dogs could become contaminated by scent-rolling.

Merigueti et al. [47] found a prevalence significantly higher in stray than in owned dogs due to a lack of anthelmintic treatments. El-Tras et al. [23], instead reported that stray habits were not significant risk factors, but fur from stray dogs presented a higher number of eggs per gram than fur from owned dogs.

A different viability of eggs was found in different studies. This high difference may be attributed to variations between environmental conditions (e.g., temperature and humidity). Although a low prevalence of embryonated infective eggs was found in the studies analyzed [22,23,37,39], unembryonated *Toxocara* eggs can develop fully on the fur under controlled conditions, as reported by Keegan and Holland [53]. In contrast, Nagy et al. [19] showed that *Toxocara* eggs on dogs’ fur did not develop to embryonated eggs, but the same authors did not exclude the contamination of the dogs’ fur with embryonated eggs from the environment.

In two studies [37,46], a significant correlation between the viability of *Toxocara* eggs on fur and breed/coat type was found; in both studies, authors reported that in dogs with short hair (as well as in puppies) there was a greater closeness of *Toxocara* eggs with the skin, where the temperature conditions are suitable for the development of the eggs.

In all of the studies, the perianal area, the back and tail resulted in sites where a higher number of *Toxocara* eggs were collected from the dog’s fur [23,37,43,44,47,48], whilst in Sowemimo and Ayanniyi [45], the neck region showed higher egg numbers. However, *Toxocara* eggs were also found in other regions such as the head, abdomen, and limbs [23,41,44,45,46], thus showing a potential zoonotic risk for humans because these sites represent areas of contact with people and other dogs. This risk could be potentially higher in AAIs, involving patients and/or users more susceptible (young or old and immunosuppressed people) to zoonotic infections [10,11,54,55]. Moreover, during AAIs, bodily contact (i.e., petting, embrace, staying close) is very important, as reported by Shen et al. [8] because it is considered a desirable behavior for patients/users, and a fundamental element for the effectiveness of the interventions themselves.

The age of dogs could not be a key risk factor, in fact, some authors showed a higher prevalence in dogs >12 months [39,40,41], whilst others in puppies, justifying the higher prevalence of *Toxocara* eggs found on the fur with correlation to the higher number of sources of infections in puppies than in older dogs [22,23,37,38,42,46,47]. Recent studies on the characteristics of dogs involved in the AAIs underline the need to consider subjects of at least one year of age that are able to express an intraspecific and interspecific assortment of behaviors, useful for the interaction with patients/users [6,27,28,56,57,58].

Although the number of studies included in this systematic review was low (only 14 papers) and their experimental designs were very heterogeneous (e.g., number of dogs involved, age, breed, length of fur, etc.), these findings may contribute to increase attention to the potential zoonotic risks related to dogs included in AAIs. However, the current scientific literature concerning potential zoonotic risks during AAIs [28,54] never refers to the hazard deriving from the presence of *Toxocara* or other helminth eggs on dogs’ fur. Therefore, a regular and complete parasitological monitoring of dogs involved in AAIs is advisable in order to prevent animal and/or human infection [59].

## 5. Conclusions

Even if the studies considered in this systematic review evidenced a low prevalence of infectious (embryonated) eggs of *Toxocara spp.* on dog’s fur, the potential zoonotic risk should not be disregarded. In addition, it is important to underline that dog hair may be contaminated with eggs of other helminths (e.g., the Taeniidae *Echinococcus multilocularis* and *Echinococcus granulosus* sensu lato) that may have a higher potential of zoonotic infection, because eggs of these species are immediately infective [19,60].

Therefore, the following veterinary actions would be advisable for dogs involved in AAIs: (1) to improve the health care surveillance through an accurate and regular parasitological monitoring not only of feces but also of dogs’ fur; (2) to monitor the activities and lifestyle of dogs (food, habitat, interaction with other dogs or other animals, attendance of dog areas in the park, rolling on grass or feces or animal carcasses) in the days and/or in the steps preceding the sessions with patient and/or user involved in AAIs; and (3) to improve hygiene procedures before and after handling and/or contact with dogs.

## Figures and Tables

**Figure 1 animals-09-00827-f001:**
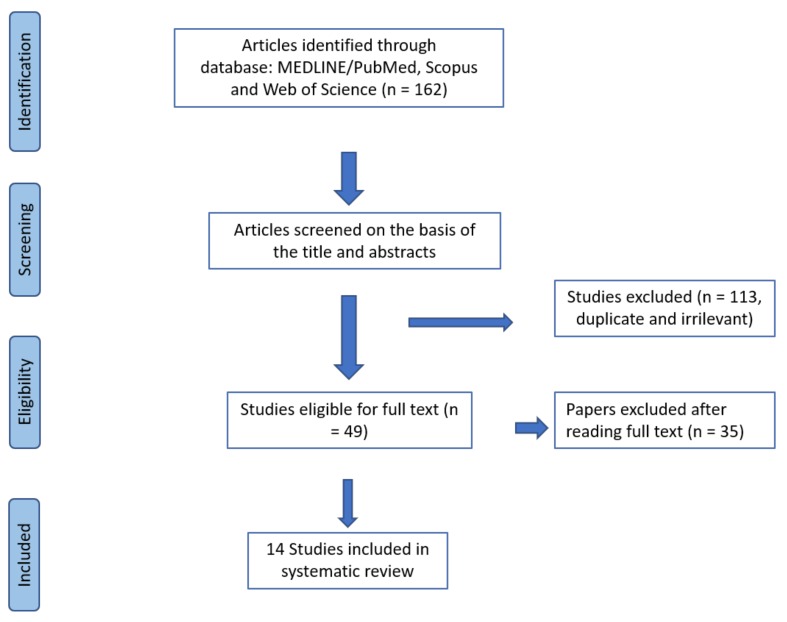
Flow diagram of the steps followed in the search strategy.

**Table 1 animals-09-00827-t001:** Characteristics of the studies concerning the presence of *Toxocara* eggs on the fur of dogs.

First Author, Year of Publication	Number of Dogs	Type of Sample	Age of Dogs	Fur Length/Breed Involved	Attitude	Prevalence of Positive Fur Dogs (%)	Body Region: Prevalence per Fur Area Sampled	Reference
Sivajothi, 2018	236	Fur and feces	<1 year 1– 6 years >6 years	Long hair coated dogs Short hair coated dogs	N.A.	60/236 (25.4)	Perianal region: 86.67%; Tail regions: 56.67%; Ventral and lateral abdomen: 51.67%; Head and Neck region: 36.67%	[44]
Merigueti, 2017	165	Fur	<1 year >1 year	Short Long	Stray Owned	11/165 (6.7)	Perianal region: 72.39%; Upper tail regions: 22.39%	[47]
Rojas, 2017	96	Fur	Young Adult Geriatric	Short (≤0.5 cm) Long (>0.5 cm)	Stray dog Not stray dog	40/96 (41.7)	Head: 14.58%; Perianal region: 20.83%; Hindquartes: 10/10.82%	[48]
Sowemimo, 2016	267	Fur and feces	0–6 months 7–12 months >12 months	Local Exotic	Free-roaming Kennel	48/267 (18.0)	Neck: 45.83%; Back: 47.91%; Anal region: 35.42%	[45]
Paoletti, 2015	676	Feces (n = 502) and Fur (n = 174)	≤12 months >12 months	N.A.	Private dogs Kenneled dogs	5/174 (2.9)	N.A.	[38]
Oge, 2014	100	Fur and feces	Puppy (<6 months) Young (6–12 months) Adult (>12 months)	N.A.	Owned dogs	14/100 (14.0)	N.A.	[43]
Tavassoli, 2012	138	Fur	Puppy (<6 months) Adult (>6 months)	Different breeds	Farm sheepdogs Pet Dogs	50/138 (36.2)	N.A.	[42]
El-Tras, 2011	120	Fur and feces (n = 100); Fur (n = 20)	Puppy < 6 months) Young (6–12 months) Adult (>1 year)	Breed and fur type according to Sato et al. [49]	Stray dogs Domestic dogs	17/64 (26.6 stray) and 6/56 (10.7 domestic)	N.A.	[23]
Keegan, 2010	182	Fur	<1 year >1 year	Short Long	Dog grooming parlor Veterinary practiceIndividual dog owner Boarding kennel	16/182 (8.8)	Head: 31.25% Neck: 25.0% Back: 43.75% Perianal region: 18.75%	[41]
Amaral, 2010	104	Fur	Puppy (<6 months) Juvenile (6–12 months) Adult (>12 months)	Short Long	Stray dogs Owned dogs	25/104 (24.0)	Perianal region: 24.0%	[46]
Overgaauw, 2009	240	Fur (n = 148) and feces (n = 92)	0.5–13 years	Short hair breed Long hair breed	N.A.	18/148 (12.2)	N.A.	[40]
Roddie, 2008	100	Fur and feces	Puppy (<6 months) Juvenile (6–12 months) Adult (>12 months)	N.A.	Stray dogs	67/100 (67.0)	N.A.	[37]
Aydenizoz-Ozkayhan, 2008	51	Fur	Puppy Young Adult	Short Medium Long	Breeds *	11/51 (21.57)	N.A.	[22]
Wolfe, 2003	60	Fur	8 weeks—15 years	N.A.	Animal care shelters Working farm dogs Domestic Pets	15/60 (25.0)	N.A.	[39]

N.A.: Not available; * Size, hair length, and coat type of the dogs were classified according to the American Kennel Club (AKC) [50] and The Kennel Club [51].

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
