# Peer review of "The Presence of Toxocara Eggs on Dog’s Fur as Potential Zoonotic Risk in Animal-Assisted Interventions: A Systematic Review"

_animals, 2019, doi:10.3390/ani9100827_

Round 1
Reviewer 1 Report
The systematic review is well done and addresses a relevant topic for animal and human health.
Reviewer 2 Report
The paper summarized and systematized available data concerning the subject of T. canis eggs in dog's fur. This zoonotic danger is worth scientific attention.
I have some technical comments and suggestions:
Inclusion criteria are too general „papers … were considered eligible if containing information related to the risk of zoonotic transmission of T. canis through the fur of dogs…..” - such information can be found also in review articles and different scientific opinions. In my opinion author have to add information that only original research studies (published or in press)were included. And that; reviews, comments, letters etc. without original data were all excluded from Systematic Review.
The list (table) of excluded papers, with information about reason of exclusion, would be needed (probably in the form of supplementary materials available only in electronic version).
Quality assessment was based on two independent person assessed included studies. It would be good to add information about method of assessment for example: the Newcastle-Ottawa Scale.
Information about searching strategy (algorithms of searching in data bases: Toxocara canis AND fur …. Etc. ) should be described in the point 2.2. not in 2.3.
Reviewer 3 Report
The review article by Maurelli et al. provides an overview of the potential risk of transmission of zoonotic Toxocara canis by contact with dogs’ fur. The authors have covered a decent range of studies in the field, however, the introduction and discussion sections needs significant improvements.
Major comments:
This review focused on zoonotic risk in animal-assistant interventions, but the authors failed to introduce the zoonotic disease of interest and its symptoms of human toxocariasis. I recommend the authors to move Line 157 – 177 in the discussion section to the “introduction section” and improve the section. Before stating the aim of the review (Line 61-63).Line 207-208: “Recent studies on the ……,the need to consider subjects with at least 1 year of age”. The authors cited some studies in reference to the statement but did not attempt to give reasons for such suggestion. The authors should provide reasons why subject with at least 1 year should be considered.
The Concluding comments are good implementable measures, however, there was little or no in depth discussion on the observed results, risk factors, public health interventions. Overall, the authors should improve the discussion section in order to enhance the quality of the manuscript.
Minor comments per heading:
Abstract:
Line 34: Remove “Results:”, and out should be “Out”Keywords:
Line 42: Limit keywords to 5 only, remove “patient, user, zoonotic risk, contact”
Results:
Line 147: Figure 1. Articles “retourned” what is that? replace with “ identified” Studies excluded (n=133, “duplicated or irrilevant” change to “duplicate and irrelevant” Studies “eligibile for full” text change to “eligible for full text”Line 149: Table 1. Add (%) in the prevalence of positive fur dogs column, that is “ Prevalence of positive fur dogs (%)” Change for example 60/236 - 25.4% to “ 60/234 (25.4%)” all through the column Paoletti, 2015, .. 5/174 – 8.7% is not correct, it should be 2.9%. El-Tras, 2011, check the values and correct them “(17-26.6% stray and 610.7%)” is not clear, I think is a mistake. Line 150: “* Not available” should be “N.A.: not available”
Discussion:
Line 154: transmission of T. canis "for" people change to “……"to" people
Reviewer 4 Report
The review article is writing well with a good idea. on the other hand, the article needs a few corrections in some words only.

Reviewer 5 Report
General comments:
Only a few, if any of the studies that are published on identification of helminth eggs on fur of dogs have confirmed the presence of Toxocara canis eggs, which can be done with molecular analyses. The manuscript should be critical with this, as the contamination of the environment can be very high with cati eggs. Even if the published papers pretend to have found T. canis eggs, it would be correct to state “eggs of Toxocara sp. “ The authors state on line 189 that „unembryonated canis egg can develop fully on the fur under controlled conditions”. This important fact should be emphasized with citations of papers documenting this. Unfortunately, the paper by Nagy et al. 2011 Berl Munch Tierarztl Wochenschr. 124:503-511 (included in PubMed) was not considered. This paper investigated the development of T. canis eggs on the fur of puppies and documented that the eggs did not fully develop to embryonated eggs in comparison with the eggs that were exposed to the dog environment. Furthermore, investigations of 291 dogs and 46 foxes revealed prevalences of 1.7 – 5.6% in dogs and much higher in foxes, but none of the eggs that they found were fully embryonated. Unfortunately, this paper is written in German and an effort is needed to study it. Furthermore, some of the citations are subdividing embryonating and embryonated eggs, but again – embryonating eggs are not infectious! In the beginning of the discussion, the paper of Overgaauw is mentioned for low risk of infection. It would be interesting to know the main facts the authors rely on (low numbers of embryonated eggs together with the fact that several hair would have to be ingested by humans, which is not a very realistic situation). Concerning the conclusions: I acknowledge that this review is of VPH significance and it is contributing to the discussion of this interesting risk factor for veterinarians and dog owners. However, I would like to suggest that the authors critically confirm the general interpretation of the infection risk with infectious (= embryonated) eggs of Toxocara spp. Furthermore, maybe it could be mentioned that other helminths such as granulosus s.l. or E. multilocularis may have a higher potential to be transmitted as these species release fully infective eggs.For example in the review of Alvarez Rojas et al. 2018: Assessing the contamination of food and the environment with Taenia and Echinococcus eggs and their zoonotic transmission. Current Clinical Microbiology Reports (2018) 5:154–163; some findings are reviewed as examples:
“Taeniid eggs can adhere to the coat of infected
dogs or foxes, and thus, there is an obvious risk originating
from direct contacts with definitive hosts. Furthermore, dogs
rolling in feces can be externally contaminated without being
infected. Investigations with five dogs infected with
hydatigena revealed 173–210 eggs/cm2 in the peri-analregion and 4–20 eggs/cm2 on other body areas, including even
13 eggs/cm2 on the nose [22]. A gravid proglottis of E.
multilocularis has been found in the peri-anal region of a
naturally infected dog [27], and examination of hair coat of
46 foxes revealed taeniid eggs in 11 animals (three cases confirmed
to be E. multilocularis) [28]. Older data document the
presence of E. granulosus eggs in dogs’ coat. Single eggs were
found in the muzzle and paws of experimentally infected dogs
[29], and taeniid eggs were detected in the anal region, around
the mouth and on the coat of rural dogs from Nigeria infected
with E. granulosus (confirmed at necropsy) [30]. These data
confirm the potential infection risk by close contact with dogs
or foxes (e.g., for hunters)”.
Round 2
Reviewer 3 Report
In the revised version of their manuscript, the authors have tried to address the deficiencies. However, the written English needs a deep review, especially the Discussion section.Line 17: “The dog” should be “Dog is the main…
Line 39-40: “it is very important attended to…..” the statement is not clear and grammatical incorrect.
Line 41, Keyword: remove comma after dog
Line 52: but “the dog” should be “dog”
In the introduction section, the authors should provide a statement on the symptoms of human toxocariasis, such information will be beneficial.
Line 108: “duplicated” should be “duplicate”
Figure 1; studies excluded (n=113, duplicated and irrilevant) should be “studies excluded (n=113, duplicate and irrelevant)”
Figure 1: Studies eligibile for full text (n=49) should be “Studies eligible for full text (n=49)”
Table 1: Remove 60/236 (25.4%) should be 60/236 (25.4), likewise others, so the authors should remove % all through, since Prevalence of positive fur dogs (%) written on the top has indicated that.
Line 180-182: the written English is not clear, rephrase.
Line 210: put comma after “In the AAls,”
Line 222: ….to express “and” should be “an” intraspecific and interspecific….
Line 234-236: these statement is not clear and not well written, difficult for readers to understand. Also, “them”???
Overall, the written English needs a deep review for the readers better understanding of the manuscript.
Reviewer 5 Report
Comments and Suggestions for Authors:
General comments: Dear authors: in my first review I stated: “Only a few, if any of the studies that are published on identification of helminth eggs on fur of dogs have confirmed the presence of Toxocara canis eggs, which can be done with molecular analyses. The manuscript should be critical with this, as the contamination of the environment can be very high with T. cati eggs. Even if the published papers pretend to have found T. canis eggs, it would be correct to state “eggs of Toxocara sp. “
This fact has not been considered nor properly addressed in your second submission. Many papers that were published in the past used the term T. canis for eggs of Toxocara spp. without exact species determination. It is important not to perpetuate such incorrect results.
Furthermore, I would suggest to prepare a short statement in the introduction concerning the diagnosis of Toxocara spp. by morphology or genetic analysis, and to mention that your review accepted contributions documenting Toxocara spp. without differentiation between T. canis and T. cati. Without fishing for citations and you do not have to cite the recently published paper by Vienazindiene, et al., “Longitudinal study for anthelmintic efficacy against intestinal helminths in naturally exposed Lithuanian village dogs: critical analysis of feasibility and limitations”. Parasitology Research 117, 1581 – 1590 (2018), but this paper clearly shows that T. canis and T. cati eggs are often excreted by dogs and only a correct diagnosis can overcome this problem.
Further suggestions:
For example line 25: do you restrict the evaluation to canis eggs or would it not be better to evaluate for Toxocara spp.? Line 76: please confirm for all these studies that canis and T. cati have been differentiated or use Toxocara sp. Line 81: If you only focus on canis, all the papers that did not strictly differentiate T. canis from T. cati should be excluded. However, this would reduce the value of your manuscript. Therefore, it would be correct to state “on the potential risk of Toxocara egg transmission through contact with fur of dogs”. Line 107: Apologies, but again – a systematic review should critically consider the state of the art methods for the detection of a parasite. It is obvious that many studies diagnosed the infection only to the genus level. Therefore, again, it would be better to state “ detect eggs of Toxocara” instead of T. canis. Legend of Table 1: in most of the papers summarized, the species was not determined and again – it is only correct if you state: “presence of Toxocara eggs”. Moreover, in the following text concerning the results: T. canis should only be used if a clear species diagnosis was performed. Sorry if I am insisting on this, but this is a relevant point. In my first review I stated: “The authors state on line 189 that „unembryonated canis egg can develop fully on the fur under controlled conditions”. This important fact should be emphasized with citations of papers documenting this.”
None of the citations that were indicated in the resubmission documented this fact. In one paper it was mentioned that eggs can develop on the fur without documenting it, and therefore this is not scientifically correct to be cited as a fact. On the other hand, the Nagy et al. 2011 paper documented in an experimental study that eggs of T. cati did not fully develop to embryonated eggs on the fur of dogs in comparison with the eggs that were exposed to the dog environment but not in contact with the dogs
